# Assessing Brain Processing Deficits Using Neuropsychological and Vision-Specific Tests for Concussion

**DOI:** 10.3390/sports12050125

**Published:** 2024-04-29

**Authors:** Brent A. Harper, Rahul Soangra

**Affiliations:** 1Department of Physical Therapy, Crean College of Health and Behavioral Sciences, Chapman University, Irvine, CA 92618, USA; soangra@chapman.edu; 2Department of Physical Therapy, Radford University, Roanoke, VA 24013, USA; 3Fowler School of Engineering, Chapman University, Orange, CA 92866, USA

**Keywords:** neuropsychological tests, post-concussion syndrome, psychomotor performance, vision screening, visual motor

## Abstract

Introduction: Since verbal memory and visual processing transpire within analogous cerebral regions, this study assessed (i) if a visual function can predict verbal memory performance. It also hypothesized whether neurocognitive (e.g., ImPACT) tests focusing on the Visual Memory and Cognitive Efficacy Index will predict Verbal Memory scores and (ii) if vision metrics and age can identify individuals with a history of concussion. Finally, it also hypothesized that King–Devick and near point of convergence scores alongside age considerations will identify candidates with a prior reported history of concussion. Materials and methods: This observational cohort assessed 25 collegiate ice hockey players prior to the competitive season considering age (19.76 ± 1.42 years) and BMI (25.9 ± 3.0 kg/cm^2^). Hypothesis 1 was assessed using a hierarchical (sequential) multiple regression analysis, assessing the predictive capacity of Visual Memory and Cognitive Efficacy Index scores in relation to Verbal Memory scores. Hypothesis 2 utilized a binomial logistic regression to determine if King–Devick and near point of convergence scores predict those with a prior history of concussion. Results: Hypothesis 1 developed two models, where Model 1 included Visual Memory as the predictor, while Model 2 added the Cognitive Efficacy Index as a predictor for verbal memory scores. Model 1 significantly explained 41% of the variance. Results from Model 2 suggest that the Cognitive Efficacy Index explained an additional 24.4%. Thus, Model 2 was interpreted where only the Cognitive Efficacy Index was a significant predictor (*p* = 0.001). For every 1 unit increase in the Cognitive Efficacy Index, Verbal Memory increased by 41.16. Hypothesis 2’s model was significant, accounting for 37.9% of the variance in those with a history of concussion. However, there were no significant unique predictors within the model as age (Wald = 1.26, *p* = 0.261), King–Devick (Wald = 2.31, *p* = 0.128), and near point of convergence (Wald = 2.43, *p* = 0.119) were not significant predictors individually. Conclusions: The conflicting findings of this study indicate that baseline data for those with a history of concussion greater than one year may not be comparable to the same metrics during acute concussion episodes. Young athletes who sustain a concussion may be able to overcompensate via the visual system. Future prospective studies with larger sample sizes are required using the proposed model’s objective metrics.

## 1. Introduction

A concussion is categorized in the broader diagnosis of mild traumatic brain injury (mTBI). Estimates within sports participants range between 1.6 and 3.8 million episodes annually, making it a public health issue [1]. Over 1.25 million high school athletes in the United States participate in sporting events annually, accruing over 60,000 concussions yearly, which makes this issue a public health concern for adolescents [2]. Unfortunately, at least 53% of those sustaining a sports concussion fail to report them [3]. Other studies estimate that symptomatic sports participants exhibit non-reporting behaviors in 50–80% of cases when examined by healthcare providers about their concussion symptoms [4,5,6]. The American College of Sports Medicine [7] purports that these estimates are too conservative and that approximately 85% are not reported. Based on these high non-reporting estimates, there is a need to identify additional assessments that might identify a previously sustained and unreported concussion. These combined assessments may also help to identify those who might have been exposed to concussive forces but who are unaware they might have sustained a concussion due to a lack of symptoms or a failure to link overt signs and symptoms to an event they deemed innocuous.

For sports concussion to identify when a concussion has occurred and when one might be ready to return to sport, the Zurich concussion consensus statement [8,9,10] supports oculomotor screens and neuropsychological (NP) assessments to identify those at risk for concussion following potential exposure to concussive forces, regardless of the reported absence or presence of symptoms. Objective neurological side effects can include impaired vision (e.g., eye movements and scanning), balance, or proprioception; delayed reaction time; and poor mental processing lasting up to weeks or months [11]. One NP assessment used to assess cognition as a screen for concussion, remove from play (RFP), and return to play (RTP) decisions is the ImPACT test (Immediate Post-Concussion Assessment and Cognitive Testing) [12]. NP tests, like ImPACT, can provide an indirect mechanism for assessing optimal functions within multiple regions and interactions within the brain. This return to sports consensus statement emphasizes the importance of using combined tests, such as simple eye function and neurocognitive tests, for making decisions.

Abnormal eye movement is the first biomarker for impaired brain function [12,13,14]. Maruta et al. studied eye movements as a potential screening tool for mTBI. They found that errors in the visual-tracking gaze occurred in individuals with mTBI and increased with increasing severity, suggesting that visual-tracking errors are a functional screen to identify mTBI and assess its severity [15].

Multiple regions of the brain, primarily in the cerebral white matter, work together to interpret information provided by the visual system [16,17,18,19]. Cerebral white matter tracks tend to be vulnerable to shearing forces, such as that associated with mTBI, potentially leading to structural damage. White matter damage can lead to deficiencies in predictive timing, attention deficits, impaired coordination of movement patterns, and decreased balance. Magnetic resonance (MR) imaging scans using diffusion tensor imaging (DTI) to detect diffuse axonal injury (DAI) in the subcortical white matter have been used to detect and quantify intrinsic microstructure and micro-dynamic tissue features as an indicator of microstructural integrity [15,19]. A majority of DTI imaging utilizes fractional anisotropy (FA) to assess the integrity of the brain’s microstructure tissue, where higher FA indicates increased connectivity and good white matter integrity and function [19]. Hulkower et al. identified that 96% of the studies they reviewed identified that those with mTBI had low FA regardless of acuity across time from acute (2 weeks), through subacute (2 weeks to 1 year), to chronic (greater than 1 year) [19]. These findings could suggest that DAI can still be identified in chronic mTBI despite individuals having been “cleared” to return to sports or life activities and considered “recovered” from the original concussive incident. Thus, it is possible chronic deficits related to cerebral processing of verbal and visual information secondary to chronic mTBI may be appreciated during the ImPACT Verbal Memory (VerbM) and Visual Memory (VisM) subtests, which were designed to measure and evaluate cerebral verbal and visual memory processing and expressed via composite scores [20]. If this is the case, ImPACT testing, in association with other clinical tests, might identify how well the brain is functioning after a concussive event.

Over the past few years, objective neurocognitive testing has become a standard for assessing the neurological deficits related to concussions for return-to-play decisions in sports [21]. The King–Devick (K-D) is a practical and commonly used rapid eye movement concussion screening test. Baseline scores are collected for each individual and compared with the same individual’s post-injury scores to determine if deficits are present, indicating that additional concussion assessments are needed [22,23]. K-D has previously been used as a baseline screening metric for athletes of different sports [24,25,26,27]. Another visual deficit screening metric for concussive episodes is near point of convergence (NPC) [28], which can even be used months after the initial injury [29]. Worse or increased scores on NPC, when compared to baseline scores, with or without concussion symptoms, indicate potential recovery limitations of the ocular–motor system [28].

Objective metrics are helpful in verifying the presence or absence of symptoms, especially considering the prevalence of non-reporting behaviors. However, assessing athletes at risk for injury should also include subjective variables such as a prior history of concussion. Recent studies have suggested an association between a history of concussion, poor performance on the K-D, deficits on the ImPACT test, or balance impairments with an increased risk of injury [14,22,30,31]. There is limited research examining the utility of a screening tool combining subjective (e.g., history of concussion) and objective metrics to assess for increased in-season injury risk. 

Once overt concussion symptoms have resolved, lingering deficits result in non-optimal neuromuscular control and the development of unconscious compensatory movement patterns, which manifest as subtle ocular or postural impairments and diminished processing [21,31]. These new patterns become habitual throughout a sports season, leading to an overall dysfunctional movement pattern with decreased mobility and stability, further predisposing an athlete to a musculoskeletal or concussive injury [31]. This construct was recently tested, and it was found that a previous concussion can predispose an athlete to an increased risk of sustaining a musculoskeletal injury [32,33]. Recent studies have linked previous concussions with an increased risk of suffering another concussive episode [34,35,36]. A few studies have shown that one or more prior concussions increase the risk for concussive injury 5.8 times [34,36]. 

Having objective metrics to identify those with a history of concussion, both reported, and non-reported, with lingering system deficits is vital to recovery and management of risk factors. Assessing the function of the athlete’s oculomotor system may be beneficial to evaluate for optimal function or residual impairments [30,37,38,39]. To date, the ImPACT test emphasizes that multiple composite scores, including verbal memory, visual memory, visual motor speed, reaction time, and impulse control, should be restored to healthy levels before clearance is given to returning to sport. The Cognitive Efficacy Index (CEI) score is a composite metric used to evaluate an individual’s cognitive function, integrating various cognitive and memory performance tests. The CEI score has had limited utility in regard to assisting the identification of optimal brain functioning; however, it may have more value when combined with ImPACT composite scores to help identify those who require further assessment by indirectly measuring one aspect of brain function without the necessity of imaging. Furthermore, simple clinical assessments might assist in identifying those with a concussion history or those exposed to such forces who were not aware they had experienced significant forces. Therefore, for this study, pre-season baseline testing was collected using a neurocognitive (e.g., ImPACT) test, the King–Devick (K-D), and near point of convergence (NPC) metrics on twenty-five male collegiate ice hockey players with and without a reported history of concussion. The aim of this study was dual: firstly, to test the hypothesis that scores from neurocognitive tests, such as ImPACT, on visual memory (VisM) and cognitive efficacy index (CEI) could predict verbal memory (VerbM) scores; secondly, to examine whether King–Devick (K-D) test scores and near point of convergence (NPC) measurements, when adjusted for age, could identify individuals with a previously reported history of concussion.

## 2. Materials and Methods

### 2.1. Participants

An observational cohort study involved twenty-five volunteer male ice hockey collegiate athletes as part of pre-season baseline performance metrics. Each participant completed and signed a written informed consent and Health Insurance Portability and Accountability Act (HIPAA) agreement to allow the researcher to use their personal health information. The university’s Institutional Review Board approved the study. To be included in the study, participants needed to have made the Division I men’s ice hockey team roster, were over 18 years of age, and were medically cleared to play by the team physician. Exclusion criteria included any musculoskeletal surgical intervention within the last three months, a concussion that occurred less than 12 months prior to the study, a history of lower extremity injury within the last six months prior to this study, any uncorrected visual problems, inner ear problems including dizziness and vertigo, or any other general medical conditions. Furthermore, the participants also completed a Physical Activity Readiness Questionnaire (PAR-Q) to determine proper activity clearance by answering seven yes/no questions regarding heart conditions, chest pain, loss of balance, musculoskeletal problems, and medication side effects. PAR-Q has been utilized in prior studies to help determine and ensure the ability of the subjects to participate during baseline data collection fully [25,27,40]. The PAR-Q was used in this study as a secondary check, as these athletes were previously medically cleared to play prior to this single baseline testing session. A researcher reviewed each PAR-Q question with each subject, and participants were excluded based on failure to obtain consent by answering “yes” to one or more PAR-Q questions or if any surgery was performed within the last three months. If they have a previous concussion, it must have been over one year since the initial injury. 

### 2.2. Procedures

Before data collection, the researcher developed a script to provide consistent verbal instructions during the examination. Prior to the baseline testing date, athletes were instructed to obtain a good night’s sleep and to avoid alcohol and caffeine consumption. Athletes had already made the team, and we believe knowing this allowed for authentic performance on these tests. Furthermore, the academic school year had not fully started, so we believe stress from school was likely not a factor during this baseline assessment. Data was collected before the initiation of competitive season practice and following an offseason rest period of approximately five months, during which time participants reported that, to their knowledge, they had not participated in at-risk sports or activities which may have resulted in sub-concussive impacts. The researcher first obtained anthropometric data, including height and weight. During the mass testing, counterbalancing was conducted by half of the subjects completing the ImPACT test first, followed by completion of the K-D and NPC, while the second half of the subjects completed the K-D and NPC in any order followed by the ImPACT test. This counterbalancing allowed for removing bias due to ordering across participants. Adequate rest was ensured between each test, and the total duration of testing did not exceed one hour. The testing environment for the ImPACT test was taken in a computer classroom setting with an empty seat between each participant to decrease distractions and was continuously monitored by a researcher. None of the ImPACT tests were flagged as invalid. All ImPACT test results were reviewed by an ImPACT-certified and trained clinician to interpret findings. Furthermore, as is standard on the ImPACT test, participants reported no history of attention deficit disorder (ADD) and attention-deficit/hyperactivity disorder (ADHD) or any other potential confounding variables within the concussion history of the ImPACT clinical report. 

### 2.3. Measures

**King–Devick Test:** The K-D test was performed according to the manufacturer’s recommended guidelines [30,41,42]. The K-D test required the participants to read a series of numbers from three separate cards as fast as possible without errors. The participants completed the test twice, and the researcher recorded the better of the two trials as the final score. The test was performed in front of a blank wall to limit visual distractions for the participants. Despite a learning effect [43], the K-D has a sensitivity of 86% and specificity of 94% and has been considered a valid, objective instrument to identify those needing further concussive workup [22,30]. 

**Near Point of Convergence (NPC) Test:** NPC is measured in centimeters (cm) from the tip of the nose to a stick with a 14-point font target. The measurement is taken when the subject reports seeing two distinct images (double-vision) or if the researcher observes deviation of the eyes from the reference object [44] performed in front of a blank wall to limit visual distractions, and one trial was performed based on prior literature [45,46]. A normal NPC score is considered 6 cm, with a larger score demonstrating greater deficits [46,47]. Furthermore, the NPC is reliable, with an intraclass correlation coefficient ranging from 0.95 to 0.98 [48].

**ImPACT Test:** The neuropsychological ImPACT is a computer test that takes approximately 30–45 min to complete and is used to gain initial measurements prior to sports participation. The initial data serves as a baseline metric on which subsequent tests can be compared should a concussion occur or be suspected. ImPACT includes an overall CEI and five subtest composite scores, which include VerbM, VisM, Visual Motor Speed, Reaction Time, and Impulse Control. ImPACT also includes a total concussion symptom score, sports position, and demographic information [20]. This study emphasized CEI, a measure of response bias, to evaluate the balance or trade-off between an individual’s speed and accuracy of responses when performing the symbol match portion of the test. For example, an individual might sacrifice speed to increase their accuracy or vice versa. VerbM and VisM are evaluated for how long it takes the subject to complete the test and how accurately they perform each test. VerbM evaluates attentional processes, learning, and memory within the verbal domain. It includes three components to calculate an overall composite score during which subjects identify words (i.e., word memory) previously shown, correctly match or pair various pictures or designs (i.e., symbol match) that were previously shown, and identify three letters in sequence. VisM evaluates visual attention, scanning, learning, and memory. It includes two components to calculate an overall composite score in which subjects need to identify previous pictures or designs (i.e., design memory) and to select one of two specific items when shown (i.e., X’s and O’s) [20]. The ImPACT test has been deemed a reliable [38,49] and valid [50,51,52,53] concussion assessment tool. However, some believe it has a ceiling effect that may diminish reliability in some of the composite scores [54]. ImPACT testing was completed in an academic computer classroom that was quiet, temperature controlled, and monitored by a researcher during testing to ensure the space remained amicable for testing. All computers faced the front in one direction, had computer glare and privacy screens, and the participants were spread out, having at least one empty seat between testers. 

### 2.4. Data Analysis

Descriptive statistics were calculated for demographic variables of age in years, height in centimeters (cm), weight in kilograms (kg), and body mass index (BMI) in kg/cm^2^. An athlete’s history of concussion was noted for each subject. A hierarchical (sequential) multiple regression was utilized to determine whether VisM scores and CEI scores predicted VerbM scores. Binomial logistic regression was performed to determine if K-D and NPC scores, considering age, predict those with a prior reported history of concussion. Statistical significance was set at *p* < 0.05, and all analysis was performed using IBM SPSS Statistics Version 27 software (International Business Machines Corp., Armonk, NY, USA).

## 3. Results

### 3.1. Demographics and Group Differences

Twenty-five collegiate male ice hockey players participated in the study. Demographic data of age (19.8 ± 1.4), height in cm (174.6 ± 3.6), weight in kg (78.6 ± 9.3), and BMI kg/cm^2^ (25.9 ± 3.0) for all participants are presented in Table 1 and baseline score for metrics analyzed between groups is presented in Table 2. During the pre-season testing, eleven of the twenty-five athletes subjectively reported a history of concussion more than one year prior. There were no missing data points, and no data were excluded, as all data were included in the analysis. In order to accurately define the two homogenous groups (e.g., history of and no history of concussion) and to attempt to control for lifetime concussion history recall bias, we utilized the reliable and valid [55,56,57] Ohio State University traumatic brain injury short form (OSU TBI) questionnaire. Although participants were unable to pinpoint specific concussion dates, the researchers reviewed the questionnaire with the participants to verify that the most recent concussive episode occurred over one year in the past. None of the participants reported being out for an extended amount of time from their prior concussion(s) or having residual symptoms after recovery. Of those who experienced a concussion, the average number of concussions was 2.1, with a range of 1 to 5 (1 concussion = 4; 2 concussions = 4; 3 concussions = 2; 4 concussions = 0; 5 concussions = 1). 

### 3.2. Hierarchical Multiple Regression

A hierarchical (sequential) multiple regression was conducted to examine whether VisM scores and CEI scores predict VerbM scores. In order for hierarchical regression models to be of value, they must meet multiple statistical assumptions, which are not easily achieved within a data set. If these assumptions are met, the model is appropriate, and the findings have potential value. Before the primary analysis, assumptions of the scale of measurement, linearity, multicollinearity, independence of residuals, normality of residuals, and homoscedasticity were checked. The scale of measurement assumption was met, as all variables used a ratio scale of measurement. The assumption of linearity was checked with a visual inspection of the matrix scatterplot between all variables. The assumption was met, as there was no indication of curvilinearity. Multicollinearity was checked with the values of Tolerance and VIF. Values of Tolerance were 0.62 for both variables, which is above 0.1, and VIF was 1.62 for both variables, which is below 10, suggesting that this assumption was met. The independence of residuals was checked with the Durbin-Watson test. The Durbin-Watson (*d* = 2.12) fell between 1.5 and 2.5, suggesting the assumption of independence of residuals was met. The normality of residuals was checked with the P-P plot. Visual inspection of the P-P plot indicated that the observation followed along the reference line; the histogram of the residuals was unimodal and normally distributed, while skewness and kurtosis (0.134 and −0.220, respectively) were between +1 and −1 with a non-significant (*p* = 0.599) Shapiro–Wilk test; suggesting that the normality of residuals assumption was met. Homoscedasticity was met as visual inspection of the residuals plot indicated a box or rectangular shape with equal variance with an even spread within the sample, and no outliers (i.e., −3 to +3) were identified, suggesting this assumption was met (Figure 1, Figure 2, Figure 3 and Figure 4). The supporting figures demonstrate that the statistical assumptions were met in order to run the analysis and to provide a meaningful predictive model. 

Model 1 included VisM as the predictor, while Model 2 added the CEI. Results from Model 1 were significant, with 41% of the variance in VerbM accounted for by VisM, *F*(1,23) = 15.8, *p* = 0.001, *R2* = 0.41. Results from Model 2 suggest that CEI explained an additional 24.4% of the variance in VerbM scores, *F-change*(1,22) = 15.3, *p* = 0.001, *R2-change* = 0.244. Thus, the effects of the independent variables were interpreted from Model 2. In this model, only CEI was a significant predictor (*p* = 0.001) for VerbM, while VisM scores were not significant (*p* = 0.135). For every 1 unit increase in CEI, VerbM increased by 41.2. Taken together, these results suggest that the CEI predicts higher levels of VerbM scores over and above VisM. Therefore, the time needed to take each section of the VerbM components of the ImPACT test may reflect a prediction that the deficits remain or that there is a new homeostasis for brain function when processing this information. Since CEI is a trade-off between speed and accuracy of responses, it may be that, in individuals with a history of concussion, the CEI might be more valuable than previously thought. 

### 3.3. Logistic Regression

It is difficult to find statistical significance when conducting logistic regression analysis because there are multiple statistical assumptions that must be met in order to run the regression. If assumptions are met and significance is found, it strongly supports the value of the model despite a small sample size and indicates the findings have potential value. Multiple regression was conducted to determine if K-D and NPC scores, considering age, predict those with a prior reported history of concussion (1 = have a history of prior concussion; 2 = no prior history of concussion). Before running the analysis, the assumptions of random sample, scale of measurement, multicollinearity, and linearity. The assumption of independence of residuals was not assessed as the dependent variable was a dichotomous variable. The assumption of random sampling was not met as the subjects did not represent a random and independent sample since they were a cohort study. The scale of measurement was met as the independent variables (e.g., age, K-D test, and NPC) were ratio variables, while the dependent variable was a categorical dichotomous variable. Multicollinearity was checked with the values of Tolerance and VIF. Values of Tolerance variables were above 0.1 (0.947–0.987), and VIF was below 10 (1.01–1.06), suggesting that this assumption was met. Linearity was assessed using the Box-Tidwell procedure. After applying a Bonferroni correction, which adjusted the alpha level to 0.017 (i.e., alpha = 0.05/3), interactions between the independent variables (e.g., age, K-D, and NPC) and their natural logs were not significant (*ps* ≥ 0.070), indicating linearity was met. Thus, the assumptions were met based on the metrics presented. Unlike the hierarchical (sequential) multiple regression, a binomial regression does not have a graphical representation to visually demonstrate the data.

The logistic regression model results indicated that the final model was significant and accounted for 37.9% of the variance in those with a history of concussion, *χ*^2^(2) = 8.32, *p* = 0.040, and *Nagelkerke R2* = 0.379. Despite the model being significant, there were no significant unique predictors within the model as age (Wald = 1.26, *p* = 0.261), K-D (Wald = 2.31, *p* = 0.128), and NPC (Wald = 2.43, *p* = 0.119) were not significant predictors individually. The odds of identifying those with a history of concussion based on age, K-D, and NPC were −0.396, 0.148, and 1.148, respectively. Despite the non-significance of the unique predictor odds values with a significant overall model, the trend suggests that if an increased sample size was obtained, one would expect the model to identify one or more significant predictors among age, K-D, and NPC. Therefore, this model suggests that the clinical assessments of NPC and K-D might be able to identify an individual with a history of concussion, which may necessitate further assessments depending on whether these findings match the participant’s self-report. 

### 3.4. Summary of Results

Using the VerbM and VisM portions of ImPACT in isolation is not typically carried out since ImPACT is typically used to evaluate the complexity interaction of brain regions during information processing. Historically, emphasis has been placed on ImPACT visual motor speed and reaction time composite scores. Despite this, the authors of this study believe there might be more value in the VerbM, VisM, and CEI metrics not currently utilized. The results of this study are in contrast to other studies performed on participants with a history of concussion. VerbM as a marker of cognitive processing was not predictive based on an individual’s ability to scan the environment from VisM but was influenced by CEI. In the ImPACT test, CEI does not provide a composite score, nor is it a clinical metric. CEI only applies to the ability to symbol match, providing an assessment of the trade-off between speed and accuracy, with a higher score being better, with scores ranging from 0.00 to 0.70 with a mean of 0.34, as suggested by ImPACT. In this study, those with and without a concussion history scored above the mean (0.44 and 0.38, respectively). An optimal VerbM raw score is 90. In this study, those with a history of concussion scored higher in VerbM (92.2) than those without a history of concussion (88.5). An increase in CEI dramatically increased the VerbM scores, which may demonstrate the ability to visually make quick decisions in the cognitive dilemma of being accurate while expedient in making such decisions (e.g., symbol matching). Age, K-D, and NPC were not significant predictors individually in the binomial regression model. What is interesting is the differences between those with a concussion history greater than one year compared to those without any concussion history. It makes sense that the longer one plays a sport, the more chance one has of experiencing a concussion (20.1 compared to 19.5 years of age). What is interesting is that those with a concussion history had better K-D scores versus those without (39.6 ± 0.8, 95%CI [39.3, 39.9] vs. 43.6 ± 1.9, 95%CI [43.4, 43.8]) and better NPC scores than those without (1.7 ± 0.2, 95%CI [1.6, 1.8] vs. 2.7 ± 0.4, 95%CI [2.5, 2.9]), which is the opposite one would expect. Those with a concussion typically have higher K-D scores (e.g., poorer performance) and longer NPC distances than those without a concussion. The findings in this study are in contrast to prior research.

## 4. Discussion

### 4.1. Eye Scanning and Cognitive Function

The first hypothesis assessed the neurocognitive (e.g., ImPACT) tests of VisM scores and CEI scores as predictive factors for VerbM scores. The intent was to assess if visual input affects more cognitive factors like VerbM and the time needed for accuracy. The results from this study did not indicate that VisM scores improved VerbM scores. What was confirmed is that the more time an individual takes to mix accuracy with speed (i.e., CEI), the better the score.

A prior concussion history increases the risk of suffering another concussion [34,35,36] or a musculoskeletal injury [32,33]. A history of one or more prior concussions increases the odds 5.8 times of sustaining another concussion [34,36]. However, a history of concussion does not result in poorer performance on the NPC [58] or the K-D test [10] in isolation. Therefore, combining both visual assessment metrics while considering age in those with and without a history of concussion might better identify those who have sustained a concussion. This led to the second question, in which age, K-D, and NPC scores were evaluated to identify differences between those with and without a prior history of concussion before the start of the competitive season.

### 4.2. Eye Assessments as Indicators of Concussion

A prior history of concussion increases the odds of suffering another concussion [34,35,36], current reports of symptoms may predispose an individual to suffer a concussion [14], and eye function is a primary indicator that the brain may not be optimally functioning after a concussion [12,13,14,15]. This study attempted to utilize objective metrics (e.g., K-D and NPC) to assess those with and without a history of concussion in conjunction with visual function in an attempt to identify those who may be at future risk of injury based on their prior concussive history. This study suggests a possible model by which to identify those with a history of concussion greater than one year.

### 4.3. Contrasting K-D Research Findings

Lawrence et al. reported that repetitive K-D performance pre- and post-exercise in adolescents with an acute concussion can predict those who experience a sports-related concussion with poorer K-D scores compared to healthy controls (52.7 ± 13.5 vs. 44.6 ± 8.9). However, K-D scores did become faster in those with an acute concussion over time from injury (between 10 and 24 days) [59]. Gold et al. identified higher K-D scores and longer processing speeds in older adults with a concussion and who were still experiencing symptoms ranging from 1 month to 2.4 years, suggesting these findings may be associated with cognitive dysfunction [60]. Echemendia et al. reported that K-D might help differentiate those with and without a concussive episode in professional ice hockey players but should not be used in isolation [24]. However, the findings of this study did not support these prior findings. Vartiainen et al. provided normative K-D scores for professional ice hockey players (mean age of 23.8 ± 5.6 ranging from 16 to 40 years of age) had a mean score of 40.4 ± 6.1 s with a range of 24.0 to 65.7 s and K-D performance were not associated with a prior concussion history [61]. In the current study, the K-D scores were better in those who have experienced a concussion, which may indicate that athletes with a concussion greater than one year tend to improve in their ability to perform this eye-scanning task, and some younger individuals may develop compensations over-time secondary to the initial concussion injury.

### 4.4. Contrasting NPC Research Findings

NPC is considered a potential indicator of a possible concussive event if ≥5 cm [44]. A systematic review in 2020 only indicated a moderate level of evidence showing that NPC does improve with vision training and that those with concussion have poorer NP acutely, at onset and up to several months than those without a concussive event [29]. Pearce et al. showed differences in NPC between concussed and non-concussed athletes (12.6 ± 9.0 vs. 1.5 ± 1.5) [48]. van Donkelaar et al. identified that NPC did not adequately discriminate between those with and without a concussion history (9.4 ± 1.6 vs. 8.4 ± 2.1 cm) [58]. Del Rossi reported that about 20% of the NPC were below the cut-off of ≥5 cm, leading to incorrect categorization, while there was no association between NPC and a concussion history despite better NPC in those with a history of concussion [62]. Furthermore, healthy controls have shown NPC to be between 1.63 and 8.18 cm [63,64]. In this study, NPC for both groups were well below the cut-off score, and those with a history of concussion were better than those without a prior concussion.

### 4.5. Prior Concussion Influence on Eye Function

A history of one or multiple concussions may lead to prolonged functional impairments or devastating long-term sequelae [65,66,67]. Standard pathological features of mTBI from direct head impacts (e.g., head-to-head contact) involve diffuse axonal injury (DAI) to the subcortical white matter. mTBI, such as in-direct concussion (e.g., forces transmitted to the head from a blow to the chest similar to whiplash injury), may also involve DAI; thus, DAI can occur with or without a direct blow to the head. White matter damage results in an inability to correctly time or anticipate sensory events, resulting in a temporal mismatch between the brain’s expectations and the actual sensory input [68]. Recognized or diagnosed concussion signs and symptoms typically resolve within 10 to 14 days [10]. Therefore, performance on metrics should be similar to or no different from those without a concussion history. Vision and visual input, during which the brain makes sense of the incoming information, are markers of optimal brain function. Abnormal eye movement is an initial marker for decreased or impaired brain function due to the overlapping regions of the brain governing eye movement [12,13,14]. If eye movement is a metric of cognitive attention function and tends to correlate with white matter integrity and the neural pathways known to carry out the cognitive function, perhaps it can be used as a screen to identify in-direct concussion in at-risk individuals, whether or not they have reported a history of concussion.

Danna-Dos-Santos et al. [69] studied participants with a long-standing history of concussion. The participants were fifty mTBI participants, including females and males, who had a prior history of concussion occurring during sports, military activities, or a car accident. These participants were older in years of age (*M* = 28.8; *SD* = 9.4), with the last concussive episode average at least seven years prior, and the number of previous concussive events per participant was two (*M* = 2.15; *SD* = 2). These authors assessed a standing task with eyes opened and closed while standing on a force plate in order to understand long-term mTBI effects. Danna-Dos-Santos et al. concluded that those who have “recovered” from their initial mTBI, sometimes years ago, have poor postural control when visual input is removed [69]. These authors speculate that if vision is removed or disturbed in a visually mediated environment, the lack of a visual stimulus might increase the risk of another injury, fall, or repeat concussion.

Caccese et al. [70] studied three groups, including 13 acutely (2 weeks to 6 months) post-concussion, 12 with a concussion history greater than one year, and a control group using their previously developed sensory reweighting paradigm to tease out activity from the proprioceptive, vestibular, and visual systems. Caccese et al. [70] suggest that alterations occur in sensorimotor processing in both acute and long-standing concussions; however, greater differences are more readily identified between acute concussions compared to healthy controls. Furthermore, Caccese et al. proposed that sensory integration compensations are more pronounced when two of the systems (i.e., vestibular and visual) continue to have less precise feedback and function or “increased noise” [70]. In such a case, it may be interpreted that the visual system is overcompensating, resulting in noise in the system, and when vision is removed, the deficits are more readily observed.

### 4.6. Limitations of the Model

There are several limitations of this study. This study only included members from a single cohort team who were young and healthy male ice hockey athletes, limiting generalizability. Participants self-reported their prior history of concussion, which was based on individual subjective recall, while playing ice hockey but were diagnosed by a medical professional. Furthermore, none of the subjects reported they were experiencing concussion symptoms, which may have biased the findings. Self-reported symptoms and reports of prior concussion are important since a majority of athletes do not report signs and symptoms consistent with having suffered a concessive episode [4,5,6,7]. Thus, this model may identify current deficits regardless of prior concussive history report, it may also identify those who may have been exposed to such forces and require further assessment who do not know they have sustained concussive forces. Despite finding a significant model using age, K-D, and NPC scores, no unique predictors demonstrated individual significance. This is likely due to the low sample size (*N* = 25), and if the sample size is increased to a sample similar in number (*N* = 3800) to Schneider et al. [14], it is likely that a unique predictor might be identified and statistically significant. In the future, we will replicate our findings in a more diverse sample and explore underlying mechanisms differentiating between the groups.

Self-reporting of concussion history is vulnerable to recall bias. To address this potential issue, the researchers used the OSU TBI questionnaire. However, the participants were poor historians, and eight of the eleven with a history of concussion had more than one prior concussion and were unable to provide specific dates of those events. This might have been due to the young age of the athletes and when the prior concussions might have occurred. The number of previous concussive events ranged from one to five, with four reporting one, four reporting two, two reporting three, and one reporting five previous concussions; however, for those with multiple concussion histories, the specific dates were not verifiable. Since the researchers could not validate the specific time points for each concussive event, the information was not able to be grouped and categorized over a range of time points. Furthermore, participants did not report any potential confounding variables, e.g., attention deficit disorder (ADD) or attention-deficit/hyperactivity disorder (ADHD), as previously described, However, self-report is subject to the accuracy of those reports.

It should be noted that the ImPACT VerbM and VisM composite scores for those with a history of concussion are within the suggested normal ranges. However, despite normal scores on these sections, there were still differences in performance on certain metrics between those with and those without a concussion history. This is a potential value of the regression model, in that despite being “normal” scores, there may still be integration issues, and this model provides another option to assess individuals with a long-standing concussion history, which would not otherwise raise suspicion from the ImPACT scores. Furthermore, despite having improved K-D and NPC performance, those with a history of concussion did have a 15% increase in their total symptom score, although symptom scores for both groups were low overall. The intent of this study is to focus on the potential utility of the regression Model. We suggest that the visual system may compensate, increasing the reliance on vision in those with a history of a concussive event and that, despite recovery, the individual with a long-standing history of concussion, void of any obvious concussion signs and symptoms, may still require further assessment. Furthermore, the work by Caccese et al. [70] and Danna-Dos-Santos et al. [69] suggest re-weighting different systems of proprioception, vestibular, and vision in those who have sustained a concussion. Further investigation of postural control system re-weighting is beyond the scope of this paper.

Despite these limitations, the intent of the study was to see if the model could identify differences in those with a long-standing history of concussion, especially when the history is nonspecific, lacking in detail, or not available, which is common in clinical practice. Therefore, having a model that can help a practicing clinician in the inherent ambiguity of clinical practice may be useful.

## 5. Conclusions

The model seems promising but requires further studies in order to verify the utility of the variables used in this model. The conflicting findings of this study may indicate that baseline data for those with a history of concussion greater than one year might not be comparable to the same metrics during acute concussion episodes. Young athletes who sustain a concussion may be able to overcompensate via the visual system. It is not clear if this possible reweighting is a benefit or a negative advantage. Future prospective studies with larger sample sizes on ice hockey players should be performed using the proposed model’s objective metrics during pre-season data collection.

## Figures and Tables

**Figure 1 sports-12-00125-f001:**
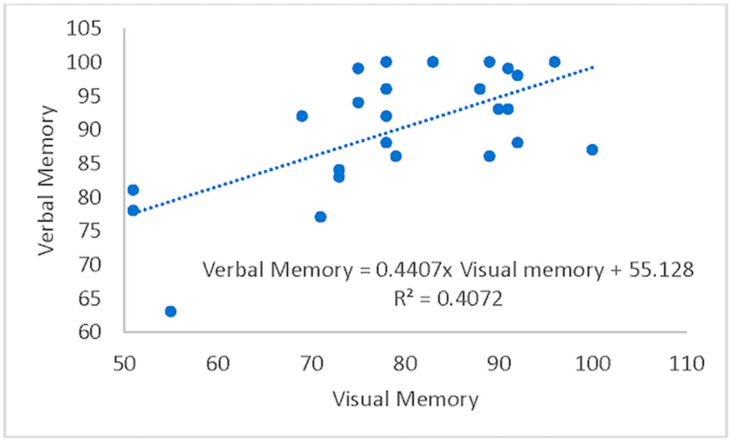
Regression plot of Verbal Memory versus Visual Memory.

**Figure 2 sports-12-00125-f002:**
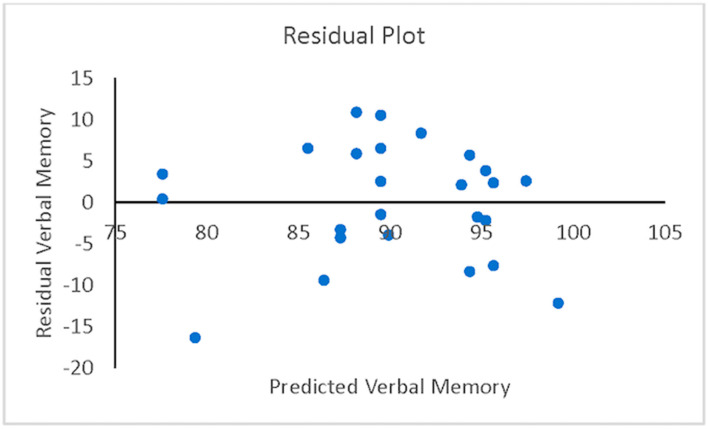
Residual plot showing residual verbal memory versus predicted verbal memory.

**Figure 3 sports-12-00125-f003:**
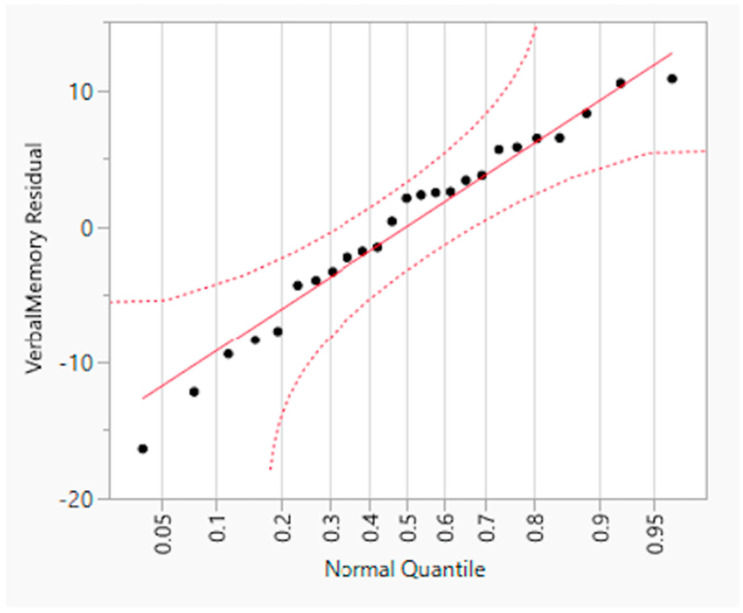
Normal Quantile Plot of residual verbal memory.

**Figure 4 sports-12-00125-f004:**
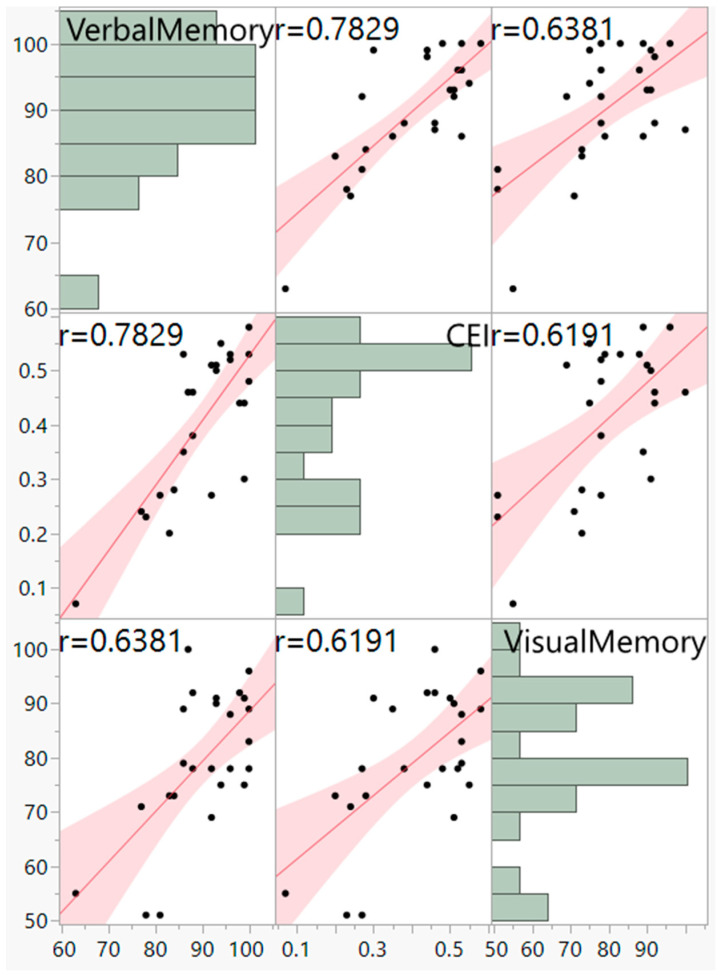
Scatterplot Matrix and pairwise correlations between verbal memory, visual memory and CEI. Demonstrates a positive moderate linear relationship.

**Table 1 sports-12-00125-t001:** Demographics of all twenty-five participants.

Anthropometric Variables	Mean ± SD	Range
Age (years)	19.8 ± 1.4	18–22
Height (cm)	174.6 ± 3.6	66–74
Weight (kg)	78.6 ± 9.3	138–211
BMI	25.9 ± 3.0	19.85–30.53

Note. Abbreviations: cm, centimeters; kg, kilograms; BMI, body mass index; values are presented as mean ± SD.

**Table 2 sports-12-00125-t002:** Baseline scores between groups categorized with and without a history of concussion.

Baseline Scores	Prior Hx of a Concussion (*n* = 11, Age 20.1 ± 0.5 Years)	No Prior Hx of a Concussion (*n* = 14, Age 19.5 ± 0.4 Years)	Cohen’s d
Visual Memory	81.7 ± 2.6	77.6 ± 4.3	1.15
Verbal Memory	92.2 ± 2.2	88.5 ± 2.8	1.47
CEI	0.44 ± 0.03	0.38 ± 0.04	1.69
K-D	39.6 ± 0.8	43.6 ± 1.9	−2.74
NPC	1.7 ± 0.2	2.7 ± 0.4	−3.16

Notes. Mean (SD); Hx = history; CEI = numeric pain rating scale; K-D = King–Devick test in seconds; NPC = near point convergence test in centimeters (cm). Values are presented as mean ± SD.

## Data Availability

The data supporting the conclusion of this article will be made available by the corresponding author upon reasonable request. The data are not publicly available due to privacy restrictions of the IRB.

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
