# Peer review of "Assessing Brain Processing Deficits Using Neuropsychological and Vision-Specific Tests for Concussion"

_sports, 2024, doi:10.3390/sports12050125_

Round 1
Reviewer 1 Report
Comments and Suggestions for Authors
Thank you for the opportunity to review the manuscript. My review focuses on the statistical analysis of the report. The study recruited 25 collegiate ice hockey players, and examined: 1) whether verbal memory performance can be predicted from visual function scores (Visual Memory and Cognitive Efficacy Index) and 2) whether history of concussion can be predicted from King-Devick and Near Point of Convergence scores, along with age. The authors used a hierarchical linear regression model and a binomial logistic regression model to answer the first and second research questions, respectively. First, I feel that the linear and logistic regression models were appropriate based on the research questions. The authors have done an exceptional job in running the models, with particular attention to the assumptions of these models. Further, all essential statistics computed in the models are reported in the manuscript. The authors also acknowledge that the small sample size was a limitation in the analysis.
Meanwhile, there is an additional analysis that, I feel, could greatly strengthen this study. Specifically, a statistical model that uses the number of concussions as an outcome variable, while using K-D test, NPC, and age as predictors/covariates, might uncover interesting findings. The potential statistical model would a Poisson/negative binomial regression model. The authors already collected data on the number of concussions from the participants. However, the binary logistic regression model used in the study resulted in the loss of information, as the model treated the outcome variable as yes (1-5 concussions altogether) or no (0 concussion). To me, K-D test and NPC scores might be influenced by the number of concussions that individuals sustained. Moreover, this outcome variable (number of concussions) is non-negative and positively skewed, with many zeros, which fits the property of a Poisson/negative binomial model.
On the other hand, if the research question is strictly about predicting yes/no of history of concussion, disregarding the number of concussions, the binary logistic regression is the only model needed. I am curious to know the thoughts from the authors on this potential, additional analysis.
Author Response
Thank you for your feedback, we have attached our response.

Reviewer 2 Report
Comments and Suggestions for Authors
The idea of this study is interesting. My recommendations are the following:
Abstract- I recommend replacing the Discussions section with Conclusions. These will specifically target the results of this study.
Introduction - I recommend to mention relevant aspects regarding the purpose of this study in correlation with the sports subjects.
Section 2.1. Participants and Procedures recommend to be mentioned distinctly as two sections.
Participants section: I recommend that table 1 be moved to this section, as well as the descriptive explanations, presented in the Results section.
I recommend mentioning the inclusion criteria of the subjects in this study.
For the Physical Activity Readiness 161 Questionnaire (PAR-Q), I recommend to mention the way of evaluating the answers and to mention α- Cronbach value, for the whole questionnaire.
Lines 291-294 recommend adding the figures mentioned.
The Discussions section - I recommend reorganizing it.
4.1. Summary of Results – I recommend moving to the Results section, without duplicating the information.
Author Response

(The authors gave the same response as above.)

Reviewer 3 Report
Comments and Suggestions for Authors
This is an interesting manuscript about an important topic. I have some comments and suggestions which may improve the quality of this paper.
Introduction
- The flow could be improved by restructuring some paragraphs and sentences. For example, the information about DTI imaging and chronic deficits (para 5-6) might fit better after discussing the brain regions involved in interpreting visual information (para 4).
- Some sentences are quite long and could be broken up for clarity (e.g. the last sentence of para 2).
- The objectives and hypotheses are not stated until the very end. Consider moving them up to give readers a roadmap of what the study will cover. The two-fold purpose is a bit unclear.
- The Introduction could be made more concise. Some information, like the details on DTI imaging and FA values, may be more appropriate for the Discussion section when interpreting results.
- Grammar and punctuation need some cleaning up, with a number of run-on sentences and missing/extra commas.
- Some key terms, like "cognitive efficacy index", are mentioned without explanation of what they are and their significance.
Methods
- Participant characteristics: While the authors mention that 25 male ice hockey collegiate athletes participated, more detailed demographic information (e.g., mean age, height, weight, BMI) would be helpful to better understand the sample.
- Inclusion/exclusion criteria: The inclusion and exclusion criteria are a bit unclear. For example, it's not specified whether participants with a concussion history were included or excluded, and if included, how long ago the concussion must have occurred.
- Testing procedures: The order of testing (ImPACT, K-D, NPC) was counterbalanced across participants, which is good. However, it's not clear if there were any rest breaks between tests or how long the entire testing session took.
- ImPACT test: More details could be provided on the specific subtests and what cognitive domains they assess. It's not clear what the "cognitive efficacy index" measures or how it's calculated.
- Data analysis: The statistical methods seem appropriate for the research questions. However, the authors don't specify what variables were entered in each step of the hierarchical regression or what the dependent variable was in the logistic regression.
- Power analysis: There is no mention of a power analysis to determine if the sample size was sufficient to detect significant effects.
- Reliability and validity: While the authors mention that the K-D and ImPACT have been shown to be reliable and valid in previous studies, they don't provide any information on the reliability or validity of the NPC test.
- Potential confounds: The authors controlled for some potential confounds (e.g., ADHD, learning disabilities) but don't mention others that could affect performance (e.g., sleep deprivation, stress, motivation).
Results
- Demographic information: While the authors provide the mean and standard deviation for age, height, weight, and BMI, it would be helpful to also include the range of values to get a sense of the variability in the sample.
- Concussion history: The authors mention that 11 out of 25 participants reported a history of concussion, but they don't provide any information on when the concussions occurred or the severity of the injuries. This information could be relevant for interpreting the results.
- Assumptions for hierarchical regression: The authors provide a detailed description of how they checked the assumptions for the hierarchical regression, which is good. However, some of the figures mentioned (e.g., Figures 1-4) are not included in the document, making it difficult to evaluate the assumptions.
- Logistic regression: The authors mention that the assumptions for logistic regression were met, but they don't provide any details on how they checked these assumptions. More information would be helpful to evaluate the appropriateness of the analysis.
- Effect sizes: The authors report the R-squared values for the hierarchical regression and the Nagelkerke R-squared for the logistic regression, which is good. However, they don't provide any standardized effect sizes (e.g., Cohen's d) to help interpret the magnitude of the differences between groups.
Discussion
- Interpretation of results: The authors provide some interpretation of the main findings, but more context is needed. For example, what does it mean that those with a concussion history had better K-D and NPC scores than those without? Is this a sign of compensation or recovery? The authors speculate about this but don't provide a clear explanation.
- Comparison to previous research: The authors do a good job of comparing their findings to previous studies on K-D and NPC in concussion. However, they don't provide much context for the ImPACT results. How do the VerbM, VisM, and CEI scores compare to normative data or previous studies in athletes with and without concussion?
- Limitations: The authors acknowledge several limitations of the study, which is good. However, they don't discuss how these limitations might impact the interpretation of the results. For example, how might the small sample size and lack of diversity affect the generalizability of the findings?
- Clinical implications: The authors suggest that the regression model could be useful for identifying athletes with a history of concussion who may need further assessment. However, they don't provide much detail on how this would work in practice. What kind of further assessment would be needed, and how would the model guide clinical decision-making?
- Future directions: The authors mention that increasing the sample size might lead to significant unique predictors in the logistic regression model. However, they don't discuss other potential future directions, such as replicating the findings in a more diverse sample or exploring the mechanisms underlying the observed differences between groups.
- Clarity and organization: Some sections of the Discussion are a bit difficult to follow. For example, the section on "Prior Concussion Influence on Eye Function" introduces some new concepts (e.g., diffuse axonal injury) that are not well-integrated with the rest of the discussion. The organization of the subsections could also be improved to provide a clearer narrative flow.
Author Response

(The authors gave the same response as above.)

Reviewer 4 Report
Comments and Suggestions for Authors
Dear Authors,
Thank you for the opportunity to review your manuscript entitled "Assessing Brain Processing Deficits Using Neuropsychological and Vision-Specific Tests for Concussion," identified as 'sports-2966988.' This manuscript is a valuable contribution to the collection "Advances in Sports Injury Prevention and Rehabilitation Strategies."
The manuscript convincingly demonstrates that the cognitive efficacy index is a significant predictor of verbal memory performance in concussion assessments. It suggests that this index could be crucial in understanding and predicting cognitive recovery or decline after a concussion. Additionally, your study underscores the challenges of using simple vision metrics and age as reliable standalone predictors for a history of concussion, highlighting the need for more robust models and further research.
Upon review, I have noted several areas where the manuscript has been significantly improved between its first and second versions, particularly in the evaluation of brain processing deficits using neuropsychological and vision-specific tests for concussions. These enhancements have deepened the analysis and interpretation of the data, making the manuscript more robust. However, there are still areas requiring further detail or adjustment:
1. The study’s sample size of 25 collegiate male ice hockey players may not provide enough power for robust statistical analysis and could limit the generalizability of the findings. Please discuss the implications of this small sample size and consider how increasing it might enhance the study's power and applicability across different sports and demographic groups.
2. The absence of a control group is a significant limitation, as it hinders the comparison of baseline characteristics and the discernment of effects specifically attributable to concussions. Discussing the inclusion of a control group comprising athletes without a history of concussion could provide a clearer contrast and strengthen the conclusions.
3. The manuscript mentions using hierarchical regression analysis but lacks detailed information about the statistical assumptions tested and any potential violations. More transparency in data handling, such as addressing missing data or outliers, would help maintain the integrity of the statistical analysis.
4. The study's cross-sectional design limits the ability to infer causality or track changes over time. Future discussions should acknowledge this limitation and explore how a longitudinal study design could offer more comprehensive insights into the progression or resolution of neuropsychological and vision-specific deficits post-concussion.
5. The study could benefit from identifying and controlling for additional confounding variables that might influence cognitive and visual outcomes, such as previous neurological conditions, medication use, or educational background.
6. While discussing the "cognitive efficacy index" as a predictor of verbal memory, the manuscript does not fully integrate how visual and cognitive tests correlate with actual cognitive deficits post-concussion. A deeper conceptual discussion of how these metrics reflect underlying neurological changes would enhance understanding and relevance.
7. There is room to expand the discussion on the limitations related to the specific tests used, particularly the sensitivity and specificity of the King-Devick and Near Point of Convergence tests at different stages of concussion recovery.
8. The conclusions should more critically reflect the limitations of the study's design and methodology. A more cautious statement of the findings, acknowledging potential biases and limitations, would prevent misleading readers about the applicability of the results.
Addressing these points would significantly enhance the scientific validity of the study and make the findings more reliable and applicable to the broader community studying sports-related concussions.
Comments on the Quality of English LanguageThe manuscript generally uses appropriate scientific language; however, careful proofreading for typographical errors, consistency in terminology, and grammar improvements would enhance readability and professionalism. Specifically, attention should be paid to the consistency of terms like "concussion" and "mTBI," ensuring they are used correctly and interchangeably only when appropriate.
Author Response
We thank you for your feedback and comments. We have addressed each item line-by-line (see attached file). Thank you for taking the time to review our manuscript.

Round 2
Reviewer 3 Report
Comments and Suggestions for Authors
The authors responded to my comments very well. Thank you.
Author Response
Reviewer 3 stated, "The authors responded to my comments very well. Thank you."
We appreciate their feedback and comments. No further revisions have been requested by Reviewer 3. Thank you.
Reviewer 4 Report
Comments and Suggestions for Authors
Dear Authors,
I express my sincere gratitude for reviewing the new version of your study. The collective efforts from you and your colleagues are reflected in the significant improvements made to Manuscript ID 'sports-2966988.' This manuscript is a valuable contribution to the collection "Advances in Sports Injury Prevention and Rehabilitation Strategies."
I commend you for diligently addressing all previous comments and meeting the outlined requests, thereby significantly enhancing the overall quality of the research. Having carefully reviewed the authors' response letter, I agree with the improvements made in this second version.
I have no additional questions and appreciate the opportunity to contribute to the review process and eagerly anticipate witnessing the continued development of your manuscript.
Thank you once again for your dedication to advancing scientific knowledge.